# Visualizing the Complexity of Knowledges to Support the Professional Development of University Teaching

**Ian M. Kinchin** [1,*] and **Paulo R. M. Correia** [2,*]

1 Surrey Institute of Education, University of Surrey, Guildford GU2 7XH, UK
2 School of Arts, Sciences and Humanities, University of São Paulo, São Paulo 03828-000, Brazil
* Correspondence: i.kinchin@surrey.ac.uk (I.M.K.); prmc@usp.br (P.R.M.C.)

**Abstract:** The idea that knowledge may exist in different forms may present a conceptual challenge for many university teachers. Our experience has shown that STEM teachers tend to view knowledge through a singular epistemological lens, driven by their disciplinary background. Such a restricted view impedes the development of teaching beyond traditional transmission models. In order to help STEM academics engage with a broader view of knowledge (and so help their students to engage in meaningful learning that does not exclude deeply held cultural perspectives), we propose a gateway into the ecology of knowledges. In this case, the gateway is created by using the analogy of protein structure—a complex idea that science teachers will be familiar with, and which demonstrates the importance of multiple perspectives on a single object. In this conceptual paper, we offer this as a tool to support the adoption of a multi-epistemic appreciation of knowledge that may lead to a more scholarly approach to university teaching.

**Keywords:** knowledge structure; abyssal thinking; consilience; teacher development; epistemology

## 1. Introduction

University academics who are embedded in the culture of their discipline will tend to see 'knowledge' through their disciplinary lens. For those with a STEM background, this will be dominated by a positivist, evidence-based and rational perspective [1]. However, when dealing with 'wicked' problems, such as climate change, sustainability or (in this case) teaching, it is apparent that rational, 'scientific' thinking is insufficient to account for the personalized and subjective knowledge that colours human understanding or the indigenous knowledges that contribute to culture and tradition. Academics are therefore encouraged to adopt an epistemologically plural stance when it comes to the 'ecology of knowledges', as described by Santos [2], that might inform teacher development and provide new concepts and new terminology to help academics articulate their teaching practice.

However, it has been shown by Skopec et al. [3] that academics working in STEM subjects have difficulty recognizing knowledge that is constructed outside of their own epistemic community. They describe this reaction against introducing ideas and narratives that might challenge the dominant view as 'epistemic fragility'. Whilst the networks of knowledge-based experts that inhabit an epistemic community might engage in intense debates, this is different from the tensions created by the acknowledgment of other epistemic communities, whose beliefs might be seen to undermine the shared beliefs of the STEM community.

In this conceptual paper, we interpret the complexity of the ecology of knowledges through an analogy from biochemistry that we feel will help academics working in STEM subjects to appreciate the plurality of knowledges and the value of the 'consilience', as developed by Wilson [4], that can be generated by acknowledging different epistemological perspectives. This consilience brings together the structured ecology of knowledge with the less familiar post-structural lens offered by a rhizomatic perspective, developed by Deleuze

and Guattari [5]. By using the analogy of protein structure (that can be simultaneously viewed through its 1°, 2°, 3° and 4° structure) as a gateway from the familiarity of the natural sciences into a new and unfamiliar perspective drawn from the social sciences, we aim to demonstrate how multiple views of knowledge can help the STEM academic to perceive a richer picture of knowledge and how it can be developed in the classroom (Figure 1).

We discuss the implications for this by considering structural descriptions of professional knowledge juxtaposed against a rhizomatic description that itself has taken its inspiration from the natural world. This will redefine expert knowledge in the context of teaching.

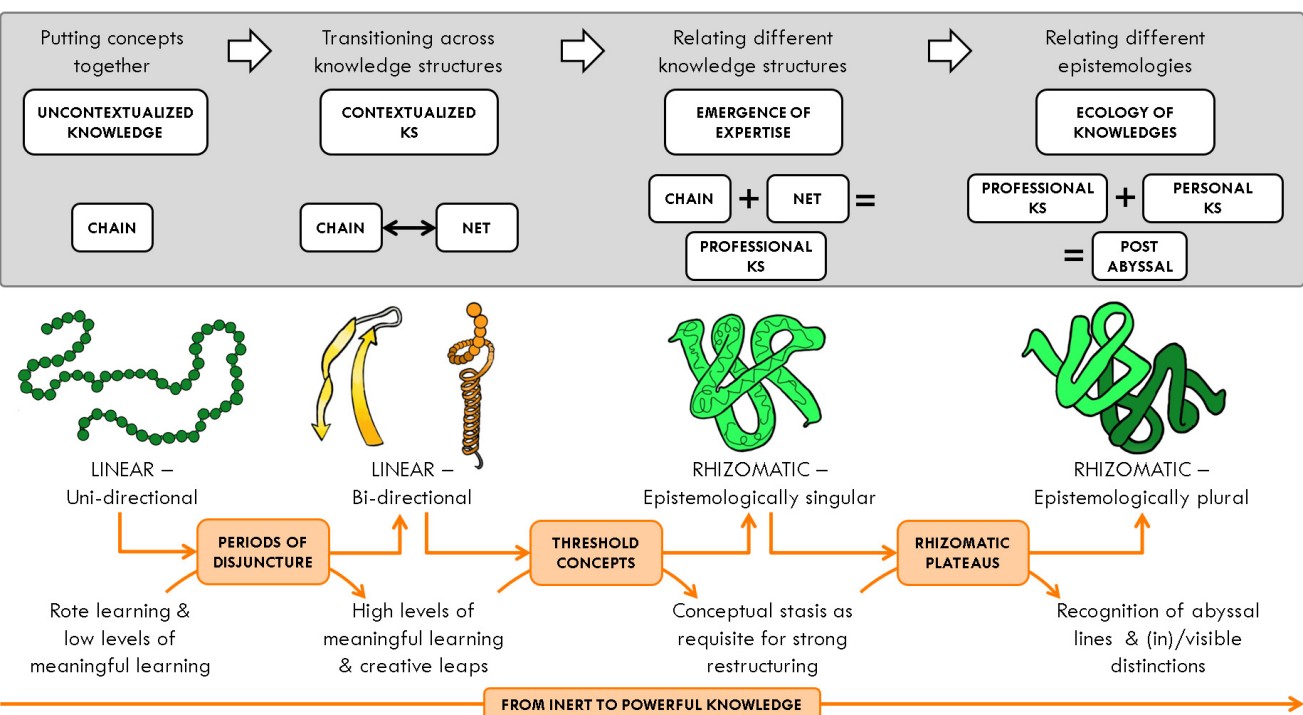

**Figure 1.** An overview of the protein structure analogy and how it can be applied to the consideration of knowledge structures (KS).

## 2. A Model from an Analogy of Protein Structure

The evolution of knowledge structures can be understood from a model with four sequential steps, presenting the emergence of expert meaning (Figure 1). Concepts are the raw material for the construction of knowledge structures. Initially isolated, they can form propositions (initial concept → linking phrase → final concept) that reveal conceptual relationships. When clearly stated, we can assess the validity of the messages communicated by propositions. Concept maps created by Novak [6] are visual organizers that depict the propositional structure of knowledge. They are useful for representing and sharing knowledge in classrooms, research teams and corporate environments [6–9].

### 2.1. An Unexpected Blackout to Contextualize Knowledge

The linear structure is the first result of articulating isolated concepts. The conceptual string is comparable to the primary structure of proteins (Figure 1), which reveals the sequence of amino acids. As with the amino acid sequence, the linear knowledge structures cannot cope with additions near the beginning of the sequence. Deletions can also disrupt the sequence, which only makes sense when reading as a whole [10]. This lack of robustness to deal with changes is a problem that keeps the knowledge uncontextualized. Likewise,

point mutations can affect the primary structure of proteins. Figure 2 shows how a discrete change in the amino acid sequence can damage the function of proteins. Sickle-cell anaemia is caused by a point mutation in hemoglobin, causing the replacement of the amino acid glutamic acid (Glu) by the amino acid valine (Val).

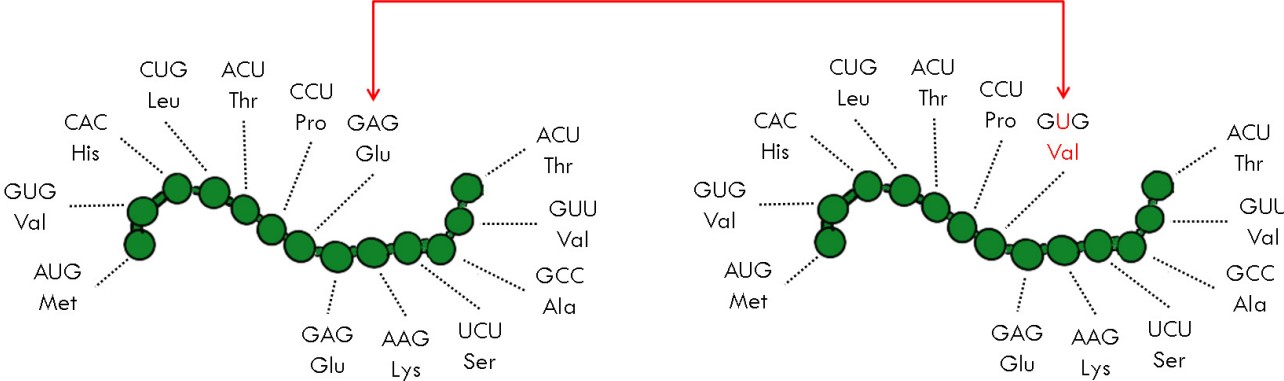

**Figure 2.** A point mutation in hemoglobin is responsible for sickle-cell anemia due to the replacement of glutamic acid (Glu) in the original sequence for valine (Val).

The limitations of linear structures become increasingly evident as learning progresses. At some point, it becomes inevitable to look for another knowledge structure more receptive to additions and deletions. Network structures allow better expression of the contextualized knowledge. As discussed in the literature [10], they can easily accommodate additions and deletions to create 'alternative routes' to connect different parts of the knowledge structure. The created links are often rich and complex, showing deep understanding.

The transition between decontextualized (linear structures) and contextualized knowledge (networks) does not follow a smooth path. It is an uncomfortable time for learners, requiring an ever-greater commitment to meaningful learning, to the detriment of rote learning. The disjunction period (Figure 3) marks the abandonment of linear structures, which prove to be useless for accommodating new information. The contextualization of knowledge requires the use of network structures. This transition is well-described by Hay et al. in the context of higher education:

> The student in this case study began a course of learning with a simple prior-knowledge structure and learnt, at first, by rote addition. Later, however, they found that what was new was irreconcilable with what they had understood to begin with. The result was a period of 'disjuncture', during which the student was less able to explain the topic than they had been before. Eventually they achieved a new grasp of meaning, but this came after a difficult period in which they might easily have given up. [10] (p. 34)

The absence of knowledge structure that marks the period of disjunction (Figure 3) causes anxiety and discomfort. It is a temporary blackout that makes us less competent than at the beginning when we had the linear structures available. This fact explains why many learners do not manage to master the topics of study as experts. The period of disjunction is the first of three transitions that we point out in our model (Figure 1). It can be compared to a dark tunnel, with the certainty that we will find a light (contextualized knowledge) at the end. Being aware of this transition increases the chances of overcoming the period of disjunction, confirming the need for a theoretical model to represent the emergence of expert meaning.

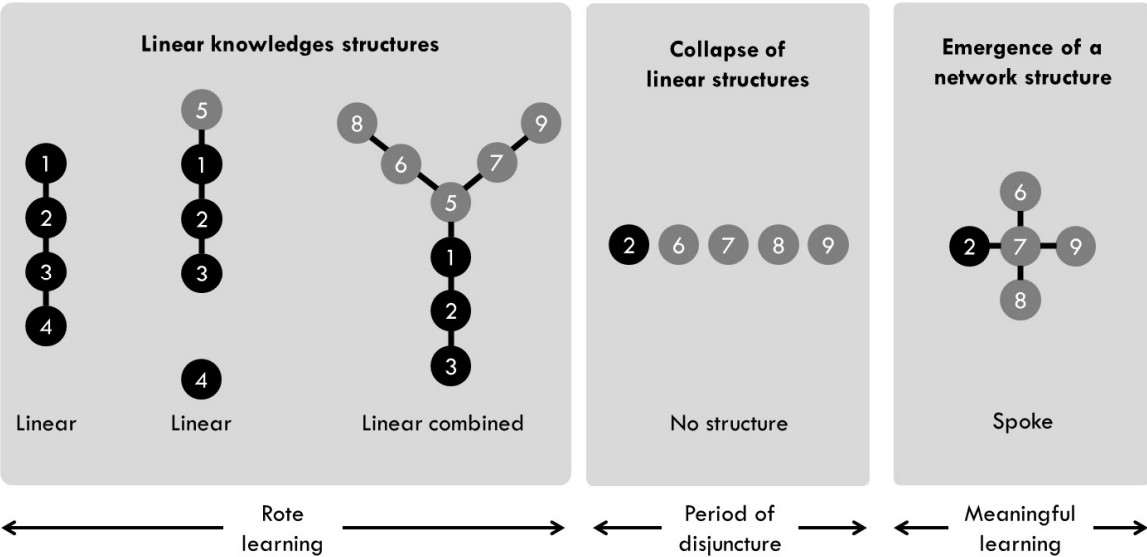

**Figure 3.** Transitioning between linear and network knowledge structures. The period of disjuncture is a temporary blackout towards the contextualized knowledge (the light at the end of the tunnel).

### 2.2. Facing Hidden Mountains to Reach the Emergence of Expertise

Contextualized knowledge facilitates the recognition of differences between linear and networked knowledge structures. While the former relates to goal-oriented chains of practice, the latter reflects conceptual understanding. In brief, we can state that while linear structures are 'active', nets are 'scholarly'. Alpha helices and beta sheets that make up the secondary structure of proteins represent this difference in the proposed model (Figure 4). These are structures typically spontaneously formed as an intermediate before the protein folds into its three-dimensional tertiary structure. Being able to recognize and use these knowledge structures is an essential competence to become an expert student [11]:

> The recognition of different knowledges has been described as essential for developing the basic characteristic of the expert student, who needs to recognize the existence and complementary purpose of different knowledge structures. [11] (p. 2)

In parallel to this student's development, it has been suggested that the emergence of the 'integrated academic'—that is, one who can work across the ecology of knowledges, oscillating repeatedly between disciplinary (science) knowledge, and teaching (social science knowledge) knowledge to support the twin activities of research and teaching—will need to adopt a multi-epistemological perspective [12].

Establishing relationships between linear and networked structures is essential to bridge the gap between theory and practice. This relationship between different types of contextualized knowledge occurs through threshold concepts, which mark the transition between contextualized knowledge and the emergence of expertise (the second of three transitions that we point out in Figure 1). They are responsible for the most significant conceptual changes. The role of threshold concepts in understanding is described by Meyer and Land as follows:

> A threshold concept represents a transformed way of understanding, or interpreting, or viewing something without which the learner cannot progress. As a consequence of comprehending a threshold concept there may thus be a transformed internal view of subject matter, subject landscape, or even world view. [ . . . ] Such a transformed view or landscape may represent how people 'think' in a particular discipline, or how they perceive, apprehend, or experience particular phenomena within that discipline (or more generally). [13] (p. xv–xvi)

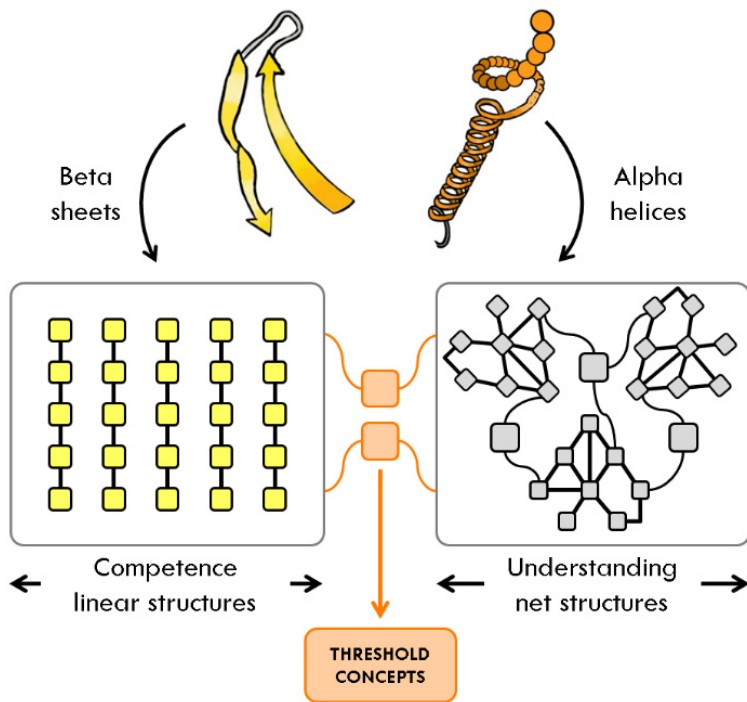

**Figure 4.** Contextualized knowledge and the differences between linear and net structures. They are comparable to beta sheets and alpha helices, respectively. Threshold concepts are hidden mountains to be overcome to connect competence and understanding.

Meyer and Land [13] offer key characteristics of threshold concepts that distinguish them from other important ideas within a discipline. Threshold concepts are likely to be:

- Transformative: they result in a change in perception of a subject and may involve a shift in values or attitudes.
- Irreversible: the resulting change is unlikely to be forgotten.
- Integrative: it 'exposes a previously hidden interrelatedness' of other concepts within the discipline.
- Bounded: it serves to define disciplinary areas or to 'define academic territories'.
- Potentially troublesome: students may have difficulty coping with the new perspective that is offered.

If the periods of disjunction are comparable to a dark tunnel with a light at the end, the threshold concepts are hidden mountains to be conquered. They suddenly appear during the learning process and can cause discomfort because no progress is apparently occurring. The punctuated learning model [14] values conceptual stasis (i.e., periods where conceptual structures appear to be unchanging) as preparation for the construction of meanings, which marks the leaps in understanding (accompanied by visible change in knowledge structures) that usually occur when we reach the 'peak of the mountain'. As Kinchin highlights:

> Stasis is required as part of the learning process: 'lining up' the segmental (beta sheets) and cumulative (alpha helices) knowledge structures for subsequent integration. [ ... ] The thresholds create moments of transformative change whilst the periods of conceptual stasis, rather than being 'nothing', are required to assemble the raw materials that will facilitate that change. [14] (p. 56–57)

Another way to explore the gap between theory and practice is from the Legitimation Code Theory (LCT), developed by Maton [15] (p. 44). His view of the arrangement of different types of knowledge, on what he refers to as the semantic plane, is formed by:

- Semantic gravity (SG), i.e., the 'degree to which meaning relates to its context' [16] (p. 129). This can be relatively stronger (+) or weaker (−) along a continuum from

theoretical to practical knowledge. Therefore, a concrete example of something tied to a particular context may be seen to exhibit a stronger semantic gravity (SG+) than a more abstract generalization derived from it (SG−).

- Semantic density (SD), i.e., 'the condensation of meaning' [16] (p. 129) that may be determined by socio-cultural practices, symbols, terms, concepts, phrases, gestures, actions, etc. Embedded within specialist texts or practices of a discipline, there are subtle meanings that are recognized by experts in the field but may be overlooked by novices who fail to appreciate the appropriate cues from what they may see as 'technically heavy' text.

The semantic plane gives an overall perspective of the role of periods of disjunction and threshold concepts in the evolution of knowledge structures (Figure 5). Uncontextualized knowledge (SG−, SD−) is transformed into contextualized knowledge after periods of disjuncture, producing practical knowledge (SG+, SD−; beta sheets) or theoretical knowledge (SG−, SD+; alpha helices). The theory–practice gap (T–P gap) is filled in when threshold concepts are acquired to bridge competence and understanding. The tertiary structure of protein represents the emergence of expertise (SG+, SD+), which combines beta sheets and alpha helices in a specific twisted shape. The three-dimensional form of a protein is key to allowing it to take part in molecular processes in the cell. In our model, this specificity is related to the professional knowledge structure, which is epistemologically singular (Figure 1), i.e., defined by the boundaries within an academic discipline.

Periods of disjuncture (dark tunnels) and threshold concepts (hidden mountains) mark the transformation of knowledge structures. Despite causing anxiety and discomfort, the degree of restructuring is different (Figure 5). Mintzes and Quinn [17] (p. 283) clarify the difference as follows:

- Weak restructuring is a gradual 'accretion' of new knowledge into an existing framework or the 'tuning' of an existing framework through the acquisition of constraining or limiting variables.
- Strong restructuring results from the introduction of powerful new organizing concepts that subsume existing ideas and forge fundamentally novel explanatory or descriptive frameworks of knowledge.

Our model captures this difference when we consider the transitions of protein structure from primary to secondary (weak, periods of disjunction), and secondary to tertiary (strong, threshold concepts). The complexification of protein structures is a valid epistemological analogy:

> Epistemologically, weak restructuring may be viewed as a kind of 'bottom-up' process characterized by incremental, cumulative, and limited change, while strong restructuring is seen as a 'top-down' process of wholesale, abrupt, and extensive modification of the cognitive structure. [17] (p. 283)

### 2.3. Going beyond the Abyss to Find the Ecology of Knowledges

The structure of professional knowledge (Figure 1) can be viewed as non-linear (or rhizomatic) in that it has no definitive start or endpoint and is in a continual state of becoming [5]. This fluid state of becoming is very different from a static state of being and has the potential to empower academics by liberating them from the struggle to attain a mythical state of excellence [18]. This allows academics to adopt an identity in which *becoming* is a legitimate descriptor for their practice. Within such a framework, rather than dividing the landscape into the 'known' and the 'unknown'—where the unknown contributes to a professional deficit—either side of the abyss (where the knowledges (sides) are epistemologically different) can both be seen to contribute to the state of becoming (Figure 6). Only now, the academic is also becoming epistemologically plural.

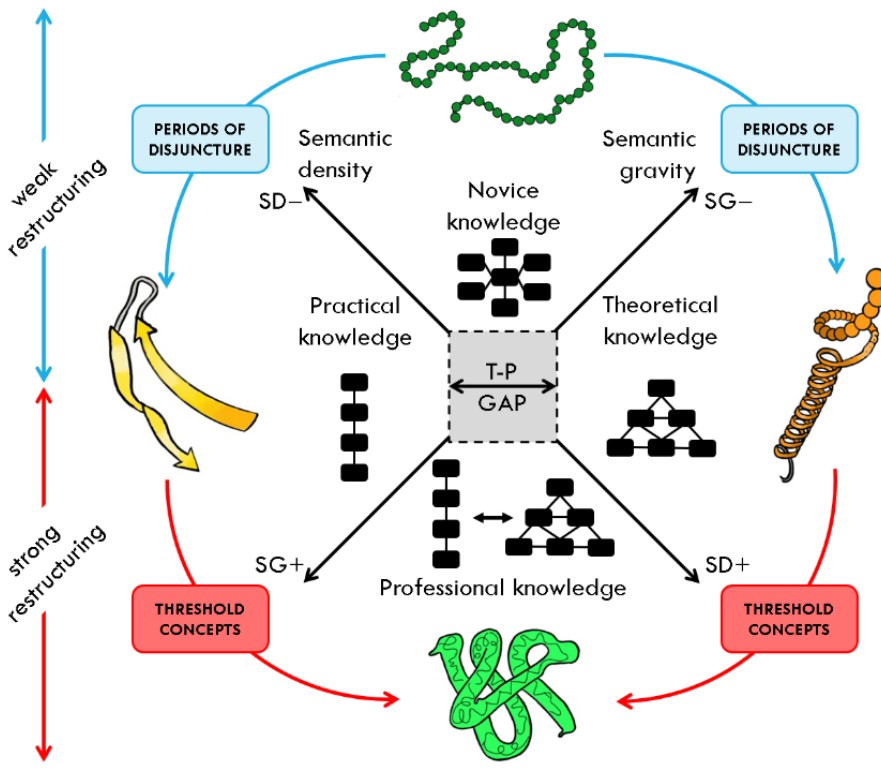

**Figure 5.** Primary and secondary protein structures represent the emergence of expertise from the semantic plane (modified from [18]). The emergence of expertise (filling the T–P gap) involves periods of disjuncture and threshold concepts to promote weak and strong restructuring, respectively.

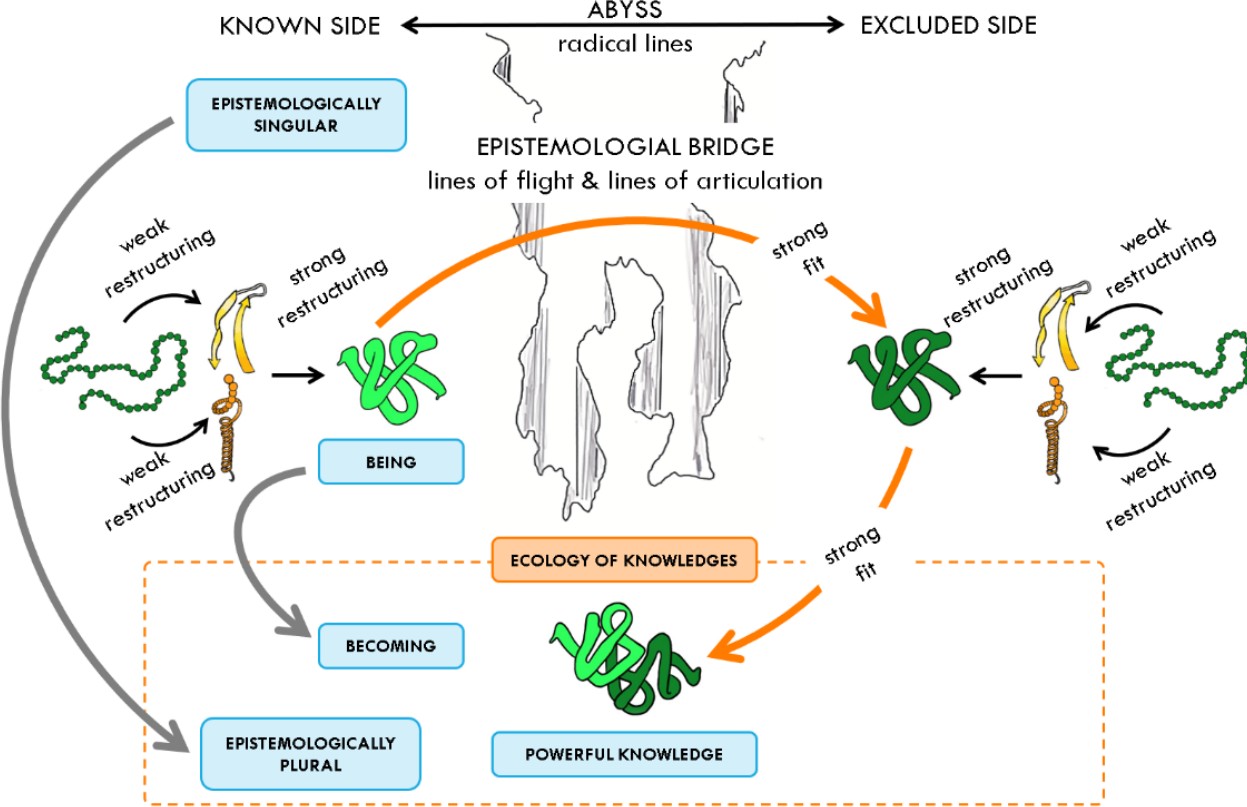

**Figure 6.** The epistemologically singular expert finds an abyss. The epistemological bridge gives access to the excluded side through a personalized experience (lines of flight and lines of articulation). The epistemological plurality is the result of the strong fit of rhizomatic knowledge structures on the opposite sides of the abyss.

The known side remains the place of relative certainties, while the unknown becomes not the 'excluded' side (sensu Santos [2]), but the 'distant' side, where becoming is possible but may take greater effort. An epistemologically singular rhizome would exclude epistemologically plural options to interpret the reality, and the discourse may be radicalized (current phenomenon). An epistemologically plural rhizome offers a route to avoid this radicalized situation and breaks down barriers between epistemic communities.

Crossing the epistemological bridge is an analogy to the fit of rhizomatic structures (tertiary structures are already formed and need to be fit to form the quaternary structure). This transition occurs through the 'strong fit' (our terminology), which involves the development of rhizomatic plateaus (the third of three transitions that we point out in Figure 1). The exploration of different epistemologies is a more intense challenge than those imposed during the previous transitions (period of disjuncture and threshold concepts). The beliefs and values created during the emergence of expertise (rhizomes) create barriers to see the other side. When it is visible, there is a difficulty in finding how to fit together different rhizomatic structures. This represents a personal journey through lines of flight and lines of articulation within the larger rhizome.

The ecology of knowledges represents an expression of post-abyssal thinking (sensu Santos [2]). It is epistemologically plural and fosters the transition from being (professional identity) to becoming (transformed/flexible identity). It also offers the highest degree of recipience (i.e., its ability to integrate new information to existing structures) compared to the other knowledge structures [19] (p. 3):

- Linear knowledge structures (chains) will exhibit 'low recipience', in that they do not support or encourage the formation of additional links to newly acquired information (i.e., reflecting a process of assimilation). This forces a period of disjunction.
- Networked knowledge structures will exhibit 'high recipience' (threshold concepts), in that they are receptive to the elaboration and are likely to support the development of new links (i.e., reflecting a process of accommodation).
- Rhizomatic knowledge structure (epistemologically singular): professional identity is formed as a result of the exploration of 'high recipience' in (b). Here, we have a period of 'professional stasis' when the academic is acting within the 'known side' of the abyss. The 'excluded side' and the existence of the abyss itself are not perceived.
- Rhizomatic knowledge structures (epistemologically plural) will exhibit the 'highest recipience' because the original rhizome becomes receptive to fit another rhizomatic structure of the excluded side of the abyss. The links to fit the rhizomes are part of a profound process of accommodation of knowledges from different academic cultures. The interdisciplinarity and transdisciplinarity involve this kind of change in knowledge structures. It is more challenging than it may appear beforehand. This may explain in part why it is difficult to implement interdisciplinary initiatives.

Inert knowledge is progressively transformed into powerful knowledge (sensu Young and Muller [20]) throughout the steps of our model (Figure 1). Powerful knowledge (that can be applied to solve problems in novel situations) is not just concerned with making links between different objects of learning but is also about the mastery of different knowledges that are brought together through weaving across the semantic plane-bringing these knowledges into contact with each other [16]. The trajectory between the novice (uncontextualized knowledge) and the expert is well described by the transitions involving periods of disjunction and threshold concepts. The focus in Higher Education is to foster the development of epistemologically singular rhizomes and the emergency of expertise.

**Author Contributions:** Conceptualization, I.M.K. and P.R.M.C.; writing—original draft preparation, I.M.K. and P.R.M.C.; writing—review and editing, I.M.K. and P.R.M.C. All authors have read and agreed to the published version of the manuscript.

**Funding:** This research received no external funding.

**Conflicts of Interest:** The authors declare no conflict of interest.

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
