# Peer review of "Visualizing the Complexity of Knowledges to Support the Professional Development of University Teaching"

_knowledge, doi:10.3390/knowledge1010006_

Round 1

Reviewer 1 Report

Very compelling and good approach. Consider to benchlearn further with systems dynamics by eg. late Jay Forrester, MIT and Dave Snowdon, Knowledge Edge as well as Latevprof Ursula Schneider ; Ignorantz Management. Clarify the conclusions more explicitly not only in abstract but also in final part

Author Response

We thank the reviewer for their comments. There could be other theoreticians whose work could be linked to this paper, and that the reader is able to make links with familiar works by other authors is helpful here so that the reader may be more able to apply this work to their own context. However, given the timeframe available for revision here, we do not think it is viable here to develop the paper in such a way, particularly as we are not familiar with the works mentioned. In addition, the development of the conclusions would not seem to fit with the conceptual paper that is being considered. We hope, therefore, that the edito will undeerstand if we do not make the suggested revisions, but reserve this further exploration for a subsequent paper. 

Reviewer 2 Report

As a reader of Deleuze I found this article very interesting and well written. It is original in describing the different epistemological approached clearly and with very helpful sketches. The argument is well developed for supporting a multi-epistemic appreciation of knowledge. It is applied to STEM teachers in the article, but it is relevant for all disciplines. I appreciated the integration of the rhizomatic perspective of Deleuze and Guattari.

I do feel however that the use of their book Difference and Repetition could strengthen the argument. The proposal of the 'ecology of knowledge' could also be developed stronger, especially in the concluding section. Here reference to Guatarri's Three Ecologies can be helpful. Otherwise this is an very good article, worthy to be published in Knowledge.

I noticed only one typo: line 251, 'beaks' should be breaks 

Author Response

We thank the reviewer for their comments. We appreciate that further exploration of the philosophical works of Deleuze and Guattari could be undertaken here (as we have done in other papers), but this paper is not really an exploration of their work, and is more about developing an approach to navigate the epistemological abyss between the sciences and the arts. As such the work by Santos is central to this and we have cited that extensively within the paper. The paper offers an epistemological bridge to STEM academics and we do not want to deter their crossing of the bridge by making this paper too philosophical in its approach.

We will amend the typo on line 251

Reviewer 3 Report

The authors provide a very original perspective of scientific teaching, using the analogy with the structure of the protein. This approach is very positive, because it escapes the positivist approach that characterizes the teaching of science and of the history of science. It is not a question of providing a teaching method, with rules and expedients, but of offering an a posteriori interpretation of a certain type of teaching. Reading this paper can be an incentive for reflection for teachers, allowing them to understand the ways in which students perceive teaching and learning. However, the authors take very specialized notions and terminologies for granted: if STEM teachers know the protein, I don't know how well informed they are about teaching methods.

Author Response

We thank the reviewer for their supportive comments. Within this paper we have to make some assumptions about the prior knowledge of the reader, in this case a familiarity with protein biochemistry. But the appriach could be applied to other aspects of science knowledge where a multiple perspectives are appreciated within one epistemic community to encourage them to revisit this with the lens offered by another epistemic community - that is, crossing the epistemological abyss.